# Imaging of Acute Abdominopelvic Pain in Pregnancy and Puerperium—Part I: Obstetric (Non-Fetal) Complications

**DOI:** 10.3390/diagnostics13182890

**Published:** 2023-09-09

**Authors:** Giacomo Bonito, Gabriele Masselli, Silvia Gigli, Paolo Ricci

**Affiliations:** 1Department of Emergency Radiology, Policlinico Umberto I Hospital, Sapienza University of Rome, Viale del Policlinico 155, 00161 Rome, Italy; giacomo.bonito@alice.it (G.B.); paolo.ricci@uniroma1.it (P.R.); 2Department of Diagnostic Imaging, Sandro Pertini Hospital, Via dei Monti Tiburtini 385, 00157 Rome, Italy; silvia.gigli@aslroma2.it; 3Department of Radiological, Oncological and Pathological Sciences, Policlinico Umberto I Hospital, Sapienza University of Rome, Viale Regina Elena 324, 00161 Rome, Italy

**Keywords:** acute abdominopelvic pain, pregnancy, postpartum, ultrasound, computed tomography, magnetic resonance imaging, obstetric complications

## Abstract

Acute abdominopelvic pain in pregnant and postpartum patients presents clinical and therapeutic challenges, often requiring quick and accurate imaging diagnosis. Ultrasound remains the primary imaging investigation. Magnetic resonance imaging (MRI) has been shown to be a powerful diagnostic tool in the setting of acute abdominal pain during pregnancy and puerperium. MRI overcomes some drawbacks of US, avoiding the ionizing radiation exposure of a computed tomography (CT) scan. Although CT is not usually appropriate in pregnant patients, it is crucial in the emergency evaluation of postpartum complications. The aim of this article is to provide radiologists with a thorough familiarity with the common and uncommon pregnancy and puerperium abdominal emergencies by illustrating their imaging appearances. The present first section will review and discuss the imaging findings for acute abdominopelvic pain of obstetric (non-fetal) etiology.

## 1. Introduction

Acute abdominopelvic pain, during and just after pregnancy, presents a diagnostic challenge, given the wide range of possible etiologies. For pregnant and postpartum women, the diagnostic approach is often more difficult, owing to several confounding factors. During pregnancy, non-specific leukocytosis, displacement of the abdominal and pelvic organs by the gravid uterus, and difficulty of physical examination can make clinical assessment challenging, especially in the third trimester [1,2]. On the other hand, the postpartum period, which may take as long as 8 weeks from birth, is also burdened by a broad spectrum of complications, significantly dependent on the delivery method: postpartum hemorrhage is the leading cause of maternal mortality world-wide [3]. Diagnostic imaging is often required to elucidate a complex clinical scenario in pregnancy and puerperium, playing a crucial role in the assessment of complications and expediting diagnosis. Ultrasound is the first imaging modality of choice both during and following pregnancy and delivery, particularly for obstetric and gynecological diseases, because of its availability, portability, and lack of ionizing radiation. However US may be limited by the small field of view, interfering structures, and body habitus, especially in the third trimester [4]. When US is indeterminate, magnetic resonance imaging (MRI) offers reproducible diagnostic imaging results that are non-operator-dependent without exposure to ionizing radiation. MRI has become a key element in the management of acute abdominal pain in pregnant and postpartum women, overcoming some of the limitations of US [5]. Diagnostic modalities employing ionizing radiation, such as computed tomography (CT), can also accurately detect many causes of abdominopelvic pain during pregnancy. A risk–benefit analysis is recommended before performing CT on a pregnant patient. However, when deemed necessary, the use of CT should not be delayed because of the concern for exposure of the fetus to ionizing radiation [6,7]. CT is more useful and plays an important role in patients with postpartum acute abdominal pain, especially after a non-diagnostic US [8]; unfortunately, the challenge of puerperium imaging interpretation lies in the wide variability of postpartum uterine and pelvic features. This article provides a review of common and uncommon gastrointestinal, hepatobiliary, urinary, gynecological, and obstetric causes of abdominopelvic pain, occurring during and postpartum, with the related imaging findings. The present instalment will discuss the main imaging features of pregnancy- and puerperium-related obstetric (non-fetal) complications.

## 2. Imaging Techniques and Safety Issues

US is widely employed as first imaging method both during pregnancy and puerperium because of its quickness, accessibility, portability, low cost, and lack of ionizing radiation. Both transabdominal and endovaginal US are commonly used to assess the uterus, the annexes, and other abdominopelvic structures. US often identifies the correct etiology of acute abdominal pain, particularly for obstetric and gynecological diseases, allowing dynamic real-time imaging. Nevertheless, US is operator-dependent and the evaluation of the bowel, ureter, pancreas, and mesenteric vasculature may be limited because of obesity, small field of view, or overlapping of other structures. Moreover, sensitivity of US decreases significantly in the third trimester of gestation due to uterus enlargement [9]. There are no documented adverse fetal effects of diagnostic US. The Food and Drug Administration proposed an upper limit of 720 mW/cm^2^ for spatial-peak temporal average intensity [10].

When US is indeterminate, cross-sectional imaging, such as MRI and CT, offers improved visualization of mother and fetus. Given the radiation exposure of the fetus with CT, MRI is the preferred modality of cross-sectional imaging in the assessment of acute abdominal pain in pregnancy [11]. The main advantage of MRI lies in the ability to depict soft-tissue deep structures in a non-operator-dependent manner, showing good overall topographic display and multiplanar imaging capabilities [12]. MRI provides additional information when further characterization is required, particularly in the setting of placental diseases such as placenta percreta and placental abruption [13]. Furthermore, MRI can play a crucial role in the diagnosis of intrauterine bleeding in pregnancy and puerperium due to its high spatial resolution and excellent sensitivity and specificity in distinguishing blood from other fluid collections. Limited availability, especially out of hours, long imaging times, and high cost are well-known challenges with MRI. Moreover, the patient is required to remain still for an extended period of time, which may be impractical in the acute setting.

There is no scientific evidence of MRI-induced teratogenesis or acoustic injuries to the human fetus and there are no specific contraindications to use MRI during any trimester of pregnancy [14,15]. MRI has been used for almost 30 years to evaluate complications in pregnancy without any demonstrated deleterious effects; however, the absolute safety of MRI in the first trimester is difficult to establish due to active organogenesis in this period [16].

The radiofrequencies employed in MRI deposit energy in the form of heat. Although the temperature increase associated with MRI is lower than the expected teratogenic level, the specific absorption rate (SAR) and tissue heating should be considered when determining which pulse sequences to use to scan pregnant women [17].

No published study in the literature compares the role of 3 T to 1.5 T in pregnancy; however, 1.5 T magnet are sufficient for clinical diagnosis and have been proven to be safe [18]. It should also be emphasized that doubling the field strength from 1.5 T to 3 T results in quadrupling of the SAR if other parameters are left unchanged [17].

Gadolinium-based contrast agents cross the placental–fetal barrier, enter the fetal circulation, and are excreted by the fetal kidneys into the amniotic fluid, where the gadolinium undergoes a time-dependent dissociation from its chelate; risks from potential exposure of the fetus to free gadolinium range from nephrogenic systemic fibrosis to stillbirth or death [14]. Gadolinium-based contrast agents (GBCAs) are classified as pregnancy Category C drugs by the FDA: their use in animal studies have shown some adverse effects on the fetus, but studies on humans are lacking. GBCAs should therefore be administered in pregnancy only when their use is crucial to establishing a diagnosis and the benefits to the patient or fetus outweigh the potential poorly understood risks [19].

According to ESUR guidelines, when there is a very strong indication for enhanced MRI, the lowest possible dose of one of the most stable agent should be administered (gadobenate dimeglumine, gadofosveset trisodium, gadoxetate disodium, gadobutrol, gadoterate meglumine, or gadoteridol) [20,21].

CT is employed more sparingly in pregnancy due to ionizing radiation, when diagnostic information required cannot be obtained with US or MRI. A thorough risk–benefit analysis is crucial before performing CT on a pregnant patient. However, the majority of diagnostic CT examinations, even performed in multiple phases, do not expose the fetus to a radiation dose that is high enough to lead to developmental/neurologic deficits or pregnancy loss [6]. This dose is lower than the recommended tissue damage threshold dose of 50 mGy (5 rad), to prevent deterministic radiation effects (i.e., effects that have a radiation limit above which they should not occur), such as fetal teratogenesis [6,22]. By contrast, there is no known threshold for stochastic effects that can occur at any radiation dose, such as radiation-induced cancer [23]. Therefore, CT can be justified in pregnancy, in selected cases, when the study is overwhelmingly in the best health interest of the mother and the patient is informed about the minimal and unknown risks to the fetus [24]; however, radiation exposure must be applied at levels as low as reasonably achievable (ALARA principle), with potential benefit outweighing the well-managed levels of risks.

Several measures such as limiting scan volume (FOV), narrowing the beam collimation, increasing pitch, and reducing milliamperage are encouraged to minimize the radiation dose without losing imaging quality. Modern scanners with dose reduction optimization can deliver much lower doses [6].

CT is the investigation of choice in the setting of life-threatening illnesses, such as hypovolemic blunt, penetrating trauma, or severe sepsis, when a variety of sites of injury or infection need to be evaluated and a prompt diagnosis is required [5]. Single CT use for speeding up diagnosis and triage of the pregnant patient has been shown to lead to favorable maternal and fetal outcomes [22].

In regards to the use of iodinated contrast media, the FDA classifies them as pregnancy category B drugs as they are considered safe in pregnant patients (with the exception of diatrizoate meglumine and diatrizoate meglumine sodium, listed as category C). No teratogenic effects have been reported with these contrast agents [25]. However, according to ESUR guidelines, iodinated contrast media should be administered only if absolutely necessary and after informed consent has been obtained; the special recommendation is to ensure that those infants exposed to iodinated contrast agents during gestation are screened for hypothyroidism during the first week [26].

Although of limited use in pregnancy, CT plays an important role in making the diagnosis or assessing the severity of peri- and postpartum complications when US findings are dubious [8]. CT is being increasingly used in the acute setting of hemorrhage, uterine rupture or dehiscence, endometritis, retained products of conception, and HELLP syndrome.

Data on the administration of iodinated or gadolinium-based contrast agents during postpartum lactation are limited; however, several studies have demonstrated that contrast media are excreted into breast milk at low levels and poorly adsorbed by the infant’s gut [27]. Moreover, the American College of Radiology states that the use of contrast media is safe in breastfeeding [28].

## 3. Obstetric Causes

### 3.1. Ectopic Pregnancy

Ectopic pregnancy (EP) is defined as the implantation of a developing blastocyst that occurs outside the endometrial lining of the uterine cavity. The incidence is 2% of all reported pregnancies. Despite advances in diagnosis and treatment, ruptured EP is the leading cause of maternal death during the first trimester, still accounting for 6% of pregnancy-related mortality [29].

The main risk factors include previous EP, history of tubal scarring related to prior pelvic inflammatory disease, previous tubal surgery, intrauterine device, and infertility treatments (such as tubal factor infertility and embryo transfer) [30]. The most common location of EP is tubal (95% of cases), with a more frequent involvement of the ampullary portion (70%) than the fimbria (11%) or the isthmus (12%). Other uncommon extrauterine implant sites include the ovary (1–3%), interstitium (2–4%), cervix (<1%), and peritoneal cavity (1.4%) [31]. Pregnancy in a rudimentary uterine horn (also called cornual) or in a previous caesarean scar are not technically EP because implantation occurs in the uterine cavity [32,33].

Clinical features are non-specific: more than half of patients experience vaginal bleeding, acute abdominal pain, and delay of an expected menses. Symptoms typically occur around 6 to 8 weeks of gestation [29]. The first diagnostic evaluation includes quantitative measurement of serum β human chorionic gonadotropin (beta hCG) and transvaginal US to confirm pregnancy. Serial β hCG level measurements are helpful to distinguish normal from abnormal gestations: a β hCG titer that rises by less than 50% in 48 h is strongly suggestive for non-viable pregnancy (GE or early pregnancy loss) [34]. When β hCG values are above a discriminatory level of 1500–2000 mU/mL, intrauterine gestational sac (GS) should be detectable on transvaginal US, with a sensitivity of 69% to 99% and a specificity of 84–99.9% [35]. The appropriate discriminatory level has been expanded recently to avoid the potential for misdiagnosis and possible interruption of an intrauterine pregnancy: therefore, ACOG put the discriminatory level at 3500 mUI/mL [29,36].

Transvaginal ultrasound shows high accuracy for the diagnosis of EP, with a sensitivity of 73–93% depending on the gestational age and the operator skills and a specificity of 84–99.9% [32,35]. US may definitively diagnose an EP, when a gestational sac (GS) with yolk sack, embryo, or both are detected in an extrauterine location (100% specificity); however, this criterion is rare, lacking in sensitivity (26%) [37,38]. The most common sonographic feature suggestive for EP is a mass separated from the ovary (60% of cases) associated with free fluid in the pouch of Douglas [32]. The mass may appear as a sac-like ring, solid or complex. On Color Doppler, a surrounding hypervascular ring (ring of fire) may be seen; however, this finding is more likely to be observed around the corpus luteum than an EP. The presence of intrabdominal fluid with floating echoes or a layering appearance is consistent with hemoperitoneum: this finding, which may be the only one, has a positive predictive value of 93% for EP, even if the differential diagnosis with the rupture of a hemorrhagic cyst must be considered [12,37]. In addition to the aforementioned operator dependence, technical issues such as interference of bowel gas and patient body habitus can limit the examination.

MRI represents a powerful problem-solving tool, especially when transvaginal US fails to detect an implantation site or differentiate EP from incomplete abortion or other acute conditions. MRI shows multiplanar capabilities, excellent soft-tissue contrast, wider scanning range, and higher sensitivity when identifying fresh blood (bloody ascites from free pelvic fluid) compared to US [39,40].

The key MRI feature of EP is an adnexal mass corresponding to the gestational sac, which appears as well-demarcated and thick-walled cystic structure separate from the ovary. The thick wall typically shows a “three rings appearance”; the outer and inner ring are thin and hypointense, whereas the middle ring is thick, displaying heterogeneous signal intensity on T1- and T2-weighted images (WI), due to intramural hemorrhages [41]. Nischio et al. observed this feature in 84–96% of cases [42]. The GS-like structure may contain non-specific fluid without solid components, resulting in hypointensity on T1WIs and hyperintensity on T2WIs, or blood, exhibiting different signal intensities based on the time of bleeding (Figure 1).

Sometimes the GS contains papillary solid components, indicating remnant of fetoplacental tissues, isointense on T2WI [38,41]. If no solid contents are depicted but only blood or fluid–fluid level on MRI, the embryocardia beats could be undetectable on US, indicating death of the embryo due to ectopic implantation. The cystic mass may exhibit heterogeneous peripheral enhancement, corresponding to the sonographic “ring of fire” sign.

Other findings include isolated hemoperitoneum, hemosalpinx and tubal wall enhancement. Hemoperitoneum, appears as fluid high signal intensity on T1WI with fat suppression and as area of heterogeneous signal intensity on T2WI. Hemosalpinx occurs after implantation of the fertilized ovum into the epithelium of the fallopian tube: a dilated tube, filled with high-signal-intensity fluid, is commonly observed on T1WI. Each of these indirect signs in patients with positive pregnancy test results and empty uterine cavity are highly suggestive of EP, even if extrauterine GS is not clearly detectable [43].

When non-contrast images are equivocal, contrast-enhanced MRI (CE-MRI) may be useful to identify the precise implantation site or to better delineate GS [38,44]. Instead, Nischio et al. observed that diagnostic accuracy was not improved significantly using non-CE and CE sequences, especially in the detection of implantation sites and recognition of GS-like structures [42]. Furthermore according to American College of Radiology appropriateness criteria for first trimester vaginal bleeding, pelvis MRI with contrast should be considered “usually non appropriate” because, if a viable pregnancy has not been conclusively ruled out, administration of gadolinium should be avoided [45].

MRI allows for the depiction of rare non-tubal forms and to distinguish between eccentric implantation in the endometrium and an interstitial ectopic pregnancy, which is hard to diagnose on US [46]. Interstitial pregnancy is a subtype of EP, resulting from implantation of the blastocyst into the intramural or interstitial portion of the fallopian tube: this condition accounts a mortality rate of 2.5%, which is seven times greater than the overall mortality rate in EP [47]. On MRI, this type of pregnancy appear as GS located in the cornual aspect of the uterine wall, and separated from uterine cavity by an intact junction zone. Owing to the proximity of this type of pregnancy to the uterine artery, rupture can lead to a life-threatening massive hemorrhage [38,47].

MRI enables also the differential diagnosis between ruptured and unruptured cases of EP before treatment. The risk of rupture increases with the enlargement of an EP. Diagnostic features of a ruptured EP include a poorly defined GS-like structure, wrapped in a hematoma in the lateral section of the uterus, associated with a great amount of hemoperitoneum. Imaging findings of tubal rupture are disruption of tubal wall enhancement and the presence of acute hematoma (low T2 signal intensity outside the implantation site) [38,41].

Corpus luteum cyst represents a potential mimic of EP. Corpus luteum originates from the ovary, whereas EP is extremely rare in the ovary. On T2WI, EP exhibits the characteristic “three ring” appearance with a thick wall, heterogeneously enhanced after administration of contrast medium, with or without papillary projections: the corpus luteum cyst, instead, typically displays thin and regular walls, with hyperintensity on T1WIs and relatively hypointense T2WIs, showing a homogeneous enhancement pattern lacking solid components [40].

Given the longer scan time compared to US, MRI should be reserved to clinically and hemodynamically stable patients, who require immediate surgical management.

In recent years advances have been made in the treatment of EP. Conservative therapies including injections of methotrexate or laparoscopic surgery (salpingostomy or salpingectomy) are widely used. However the diagnosis of EP must be established and the morphology of the GS and of the affected tube must be assessed before treatment. For a ruptured EP, emergency laparotomy is needed [29].

### 3.2. Placental Abruption

Placental abruption (PA) is defined as the premature separation of a normally implanted placenta from the underlying myometrium. PA occurs in 0.6–1% of pregnancies, with the highest reported incidence from the 24th to 26th weeks of gestation. Severe cases of PA can lead to maternal disseminated intravascular coagulation or uncontrolled blood loss, fetal demise, preterm labor, and neonatal death [48,49,50].

Risk factors for PA include hypertensive disorders, advanced maternal age, multiparity, prior PA, thrombophilia, smoking, cocaine use, premature rupture of the membranes, and abdominal trauma.

Clinical presentation and severity vary widely from totally asymptomatic cases to those where there are major fetal and maternal complications, depending on the location and the degree of PA. The main clinical features of PA are antepartum vaginal bleeding (13–25% of cases), abdominal pain, uterine tenderness, and signs of fetal distress [50,51,52].

Imaging appearance of PA can be classified according to the predominant location of the hematoma, which form as a sequela of abruption.

Retroplacental hematomas, which account for 43% of hematomas, are located between the basal plate and the myometrium and lift the placental parenchyma toward the amniotic cavity: the source of bleeding is usually from small arterioles. Retroplacental hematoma with a size greater than 50 mL or striping at least 50% of placenta from endometrium, is associated with poor fetal prognosis. Marginal subchorionic hematoma is the most frequent (57% of cases), collecting between the basal plate and the chorion [52,53,54].

A collection of blood located anterior to the placenta between the chorionic membrane and the villous chorion and circumscribed from the umbilical cord is defined as preplacental or subamniotic hematoma (Figure 2).

Bleeding in the intervillous space of the placenta is consistent with intraplacental hematoma: this uncommon condition carries a higher risk of maternal and fetal adverse events than retroplacental abruption [51,52,53].

PA can be also distinguished on the basis of the presence or absence of vaginal bleeding, being revealed versus concealed. Revealed abruption occurs when blood tracks between the membranes and the decidua, escaping through the cervix into the vagina, whereas concealed abruption consists of a blood collection behind the placenta with no external bleeding [48].

US is still the first modality of choice for placental assessment, because of its safety and availability. However the overall diagnostic performance of US in the detection of PA is poor, with reported sensitivity of less than 25% and a negative predictive value from 14 to 53% [55,56,57]. The US appearance of PA correlates with the size and location of the bleeding, as well as the time elapsed between abruption and the examination. Among 25–50% of hematomas, mostly retroplacental, remain undetected, both because acute and subacute bleeding can be isoechoic to placental tissue and because of their small size. Additionally, blood resulting from PA may drain through the cervix rather than collecting around the placenta at the time the US is performed [49,50]. Despite the aforementioned limitations, ultrasound findings, when recognizable, are highly specific for PA (92–96%), including the detection of retroplacental/preplacental collections, evidence of marginal subchorionic or intra-amniotic hematomas, or visualization of a blood clot. Moreover, a thickened placenta (greater than 5 cm) with rounded edges and heterogenous echotexture could be observed. The placenta “jiggles” when pressure is suddenly applied with the probe, the so called “jello effect”. It must be pointed out that as lesions depicted by US are relatively large, ultrasound-diagnosed PAs are associated with worse fetal outcome. Given that US is not sensitive for detection of PA, this complication must be suspected, regardless of a negative sonographic result [4,49,50,51,52,53].

Masselli et al. showed that MRI could be a useful tool for identifying PA, with high soft-tissue contrast, wide field of view, and excellent interobserver agreement [49]. The authors observed that the diffusion and T1-weighted images have higher sensitivity and diagnostic accuracy (sensitivity, 100% and 94%, respectively; diagnostic accuracy, 100% and 97%, respectively) than the T2-weighted half-Fourier RARE (sensitivity, 94%; diagnostic accuracy, 87%) and true FISP sequences (sensitivity, 79%; diagnostic accuracy 90%) in detecting hematomas. Moreover, according to changes in signal intensity of hemoglobin in placental tissue on T1 and T2WI, MRI is able to estimate the age of bleeding (Figure 3) [49].

Hyperacute hemorrhage is typically hyperintense on T2-weighted and DW images, being intermediate on T1-weighted images. Acute hemorrhage shows a drop of signal intensity on T2WI, resulting in a hyperintense T1WI. Subacute hemorrhage is hyperintense on T1WI due to the paramagnetic effect of methemoglobin. Chronic bleeding is hypointense on T1WI and T2WI. The finding of a hyperacute or acute placental hematoma should warn of the risk of progression to a higher degree of abruption, whereas subacute or late bleeding is usually stable [13,49,51]. Therefore, an accurate and timely diagnosis of PA and the prediction of its worsening are crucial when considering conservative treatment. A potentially unstable patient requires continuous monitoring and emergency preparedness. Since the diagnosis of PA is based on clinical features and not on imaging findings, MRI is not routinely performed; however, this method is extremely accurate for placental assessment, identifying the cause of second- and third-trimester uterine bleeding. MRI should therefore be considered after negative US, especially if the diagnosis of abruption could change management [13,54].

PA is often associated with fetal acidosis and hypoxia, leading to placental insufficiency and ischemia: in this regard, various studies have already shown that a reduced ADC value is a marker of placental dysfunction, which consists of a decrease in the placental surface area available for oxygen exchange and nutrient supply to the fetus [55,56]. A recent retrospective case-control study investigated whether PA, without fetal distress, could be assessed by ADC values. The authors demonstrated that ADC values on the lesions above the PA site were significantly reduced compared to those in the control group. Moreover, in the abruption group, ADC values at the abruption site were also significantly lower than in the non-abruption site within the same placenta [57].

In the management of trauma, CT is usually performed for placental and maternal injury assessment, with a reported sensitivity of 100%; however, specificity ranges from 56% to 86% depending on the operator’s skills and knowledge of the normal appearance of the placenta [7,58]. Jha et al. supported these data, also highlighting a low inter-observer agreement even among experienced radiologists; therefore, rigorous training of radiologists in detecting PA on CT is required [58].

On contrast-enhanced CT, PA is characterized by a partial or full-thickness area of low attenuation that typically forms acute angles with the myometrium. A major finding of abruption is placental non-enhancement, associated with higher risk of fetal demise: Saphier et al. proposed a CT grading system for the presence or absence of a PA, based upon the percentage of placental enhancement in pregnant patients, who underwent CT after trauma [59]. On non-contrast CT, the hematoma may be undetectable, having the same attenuation as the placenta. Hyperdense amniotic fluid from placental bleeding into the amniotic cavity may occasionally be seen. A risk–benefit analysis should be carried out before irradiating a pregnant woman. However in life-threatening conditions, CT scan should be performed, as in a non-pregnant patient [60].

### 3.3. Placental Accreta Spectrum Disorders

Placenta accreta spectrum (PAS) disorders are complex obstetric complications characterized by abnormal adhesion of the placenta, occurring when a defect of the decidua basalis allows the invasion of trophoblastic tissue into the myometrium [61]. PAS disorders are classified into three entities, according to the depth of invasion: placenta accreta (the villi simply adhere to the myometrium), placenta increta (the villi penetrate the myometrium), and placenta percreta (the villi invade the full thickness of the myometrium, uterine serosa, and often the surrounding organs) [13,61]. The incidence of PAS disorders has been increased to 3 per 1000 deliveries in the last decade, presumably due to rising use of caesarean section and other uterine surgery, which are the main risk factors [62]. Prenatal unsuspected PAS disorders are often associated with massive obstetric hemorrhage at the time of placental separation from the uterine wall, remaining the leading cause of peripartum hysterectomy in Western countries: placenta percreta (PP) may cause uterine rupture and is the most life-threatening invasion type of PAS [13,61,63]. Therefore, accurate antenatal diagnosis and identification of type of PAS disorders are crucial to plan the appropriate management with a multidisciplinary team, preventing maternal morbidity and mortality.

Pelvic US, by transabdominal and transvaginal approaches, is the recommended first-line imaging method to diagnose PAS [64,65]. Sonographic features include loss of the clear zone, defined as loss or irregularity of the hypoechoic plane in the myometrium under the placental bed, supposed to represent an abnormal extension of the placental villi through the decidua basalis into the myometrium; numerous, large, irregular sonolucent intraplacental spaces, giving the placenta a “moth-eaten” appearance and containing turbulent flow (placental lacunae), which is the most sensitive US finding in detecting all types of PAS (sensitivity 100%); myometrial thinning <1 mm; interruption of the hyperechoic uterine serosa–bladder interface; placental bulge; and a focal exophytic mass of placental tissue extending beyond the uterine serosa [4,63,66].

A prospective cohort study by Comstock et al. reported that the sensitivity and specificity of grey-scale imaging alone are greater than 90% with high predictive negative value when performed by a skilled operator [67].

Although interobserver agreement is good to excellent among experienced operators regarding the diagnostic accuracy of the individual findings, such as placental lacunae [68], others features are artifacts (myometrial thickness) that result from myometrial scars due to a previous caesarean delivery, or are rarely reported (placental bulge and focal exophytic mass) [66].

Power and Color Doppler can be helpful in establishing a differential diagnosis between placenta accreta and percreta, depicting areas of increased vascularity, with dilated blood vessels that pass through the placenta and the uterine wall, running perpendicular to myometrium and involving the uterine serosa–bladder junction; however, these techniques require more skills and experience than grey-scale imaging machines. A meta-analysis by Jauniaux et al., evaluating the diagnostic accuracy of US techniques (including grey-scale imaging, CD and PD) in women with placenta previa or with a previous history of caesarean delivery, reported a pooled sensitivity of 88% (95% CI, 81–93) and 97% (95% CI, 93–99) in retrospective and prospective studies, respectively [69]. Use of a combination of signs increases the detection rate of US for PAS disorders, especially for placenta percreta [65,69].

The disadvantages of US, including operator dependence and limited penetration/field of view, are overcome by MRI. The most attractive advantage of MRI lies in the ability to depict the entire placental–myometrial interface in detail, due to higher contrast resolution and tissue-specific characterization than US [51]. US and MRI have similar accuracy in PAS disorder diagnosis [64,70], whereas FIGO recommendations stated that MRI is “not essential” [18]; however, if US findings are equivocal for abnormal placentation, MRI represents a useful adjunctive diagnostic tool, especially for placentas in lateral and posterior positions (or in patients with previous uterine surgery), with an excellent interobserver agreement in detecting the presence and depth of placental invasion [71]. A meta-analysis including twenty studies (1080 pregnancies scanned on MRI) showed a sensitivity of 94.4%, 100%, and 86.5% for detection of placenta accreta, increta, and percreta, respectively, with corresponding values of specificity of 98.8%, 97.3%, and 96.8% [72]; however, the majority of these patients had an MRI performed after US prediction for PAS disorders. Prenatal MRI can also assess the precise topography of placental invasion and adjacent organ involvement, necessary for surgical planning [71].

MRI findings of PAS disorders that reached strong consensus recommendation (at least 80% agreement in favor) in the recent SAR-ESUR publication [73] are as follows.

Dark placental bands on T2WI: linear or nodular hypointense bands thicker (>1 cm) than the normal placental septa, extending across the myometrium–placenta interface, with random distribution, reflect increased fibrin deposition due to placental hemorrhage or infarct. Familiari et al. observed that the detection of T2 dark placental bands is the most sensitive MRI finding for the diagnosis of placenta percreta (82.6%); however, specificity was only moderate (58.5%) [72]. Therefore, this criterion is most valuable in conjunction with other supporting features. The maximum length of T2 dark bands has been shown to predict intraoperative hemorrhage [74]. Pain et al. highlighted the importance of this sign for the differential diagnosis of placenta percreta (PP) versus placenta accreta (PA) [75].

Thinning or loss of T2 hypointense interface: loss of a thin dark line between the placenta and the myometrium, depicted on T2WI [73], reflects the loss of the retroplacental clear zone on US [63].

Myometrial thinning has been described as the earliest MRI finding to suggest placenta accreta. The myometrium may appear thin (less than 1 mm) or even imperceptible in the area of placental implantation. Due to the physiological thinning of the myometrium that occurs with the progression of gestation, this sign has low sensitivity and specificity; therefore, it should not be used as an independent sign but used in conjunction with other findings suggestive for PAS [76].

Abnormal vascularization of placental bed: the vascular architecture at the placental bed appears bizarre and disorganized, with prominent and tortuous vessels non-uniformly distributed and very heterogeneous in size. These vessels may extent from the placenta to the underlying myometrium into the uterine serosa or urinary bladder wall (the so called “bridging vascularity”) [76]. The more invasive the placentation, the more pronounced the uteroplacental vascular abnormalities. This MRI criterion showed the greatest diagnostic accuracy for PAS with sensitivity, specificity, PPV, and NPV of 81.6%, 100%, 100%, and 61.1%, respectively [77]. Moreover, Chen et al. observed that abnormal vascularization of the placental bed is a specific MRI feature for differentiating PP from PA [78].

Bladder wall interruption: this sign describes disruption of the bladder wall contour and abnormal superior tenting on the bladder dome by the placenta in the supine position. The finding of placental tissue spreading into the bladder lumen is extremely specific (100%) for bladder involvement; however, it is reported in a small number of women with PP. The bladder vessel sign, defined as an abnormal vascular network within the vesicouterine space, represents an accurate predictor of bladder invasion [73,77].

Focal exophytic mass: placental tissue typically located toward the bladder or laterally toward the parametrium is very specific for PP [72].

Placental/uterine bulge: deviation of the uterine serosa from the expected plane caused by abnormal bulge of placental tissue toward surrounding organs (typically toward the parametrium and bladder) can cause the uterus to take on an “hourglass” configuration due to widening of the lower uterine segment, resulting in a loss of the typical inverted pear-shape, best depicted on sagittal and/or coronal images. This finding showed a sensitivity and specificity for the diagnosis of PAS of 76.6% and 62.5% [72] (Figure 4). The specificity of this criterion increases where the bulging in the uterine contour is associated with a focal interruption of the myometrium [73].

The others four signs that did not achieve consensus (with less than 80% agreement) were classified as “uncertain”: placental protrusion into the cervix, placental ischemic infarction, placental heterogeneity, and abnormal intraplacental vascularity [73].

Do et al. showed that radiomics features on placental MRI may represent a quantitative tool for the objective assessment of PAS severity, discriminating patients who required caesarean hysterectomy from those who did not [79].

### 3.4. Uterine Rupture

Uterine rupture (UR) is a rare yet sometimes fatal complication for both mother and fetus, often occurring during the third trimester of gestation, labor, or immediately after delivery [80].

UR is defined as a full-thickness tear of the uterine layers, including the overlying serosa, resulting in direct communication between the amniotic and peritoneal cavities. This may lead to severe uterine bleeding, fetal distress, and expulsion of the placenta and/or fetus into the abdominal cavity [81]. The average incidence reported in an unscarred uterus is 5 in 10,000 patients, whereas this rate increases to 20–80/10,000 in women with uterine scars, mostly resulting from prior caesarean section (CS). Therefore, previous CS represents the leading cause of overall uterine ruptures [82].

Uterine dehiscence (UD) refers to an incomplete separation of the myometrium at the site of a previous scar that preserves peritoneal serosa and amniotic membranes, allowing visibility of the fetus through the perimetrium: this is a much more frequent and often asymptomatic condition, which rarely results in life-threatening complications [83]. The incidence of uterine dehiscence is unknown, because it may not be recognized in the setting of a successful vaginal delivery without assessment of the lower uterine segment [84].

There is no reliable modality to predict UR or UD in patients undergoing trial of labor after caesarean section (TOLAC); however, US in the third trimester may be a useful complementary tool to predict uterine scar defects in these women [85]. Several studies have demonstrated that ultrasonographic measurements of both the full low uterine segment (LUS) and myometrial thickness during the third trimester of pregnancy are inversely correlated with uterine scar rupture/dehiscence at delivery [85,86]. US finding of thicknesses of the full low uterine segment and myometrial layer of less than 2.3 mm and 2.5 mm, respectively, could predict these complications. However, no precise threshold value could be recommended [81,85,87].

In peripartum and during delivery, the acute clinical presentation of UR obviates the need for diagnostic imaging, which is most helpful when this complication occurs earlier in the pregnancy. US is the first method of choice, owing to its availability, cost effectiveness, and lack of ionizing radiation. While transabdominal US provides a wider field of view (FOV) and better general evaluation of the pregnant status, transvaginal US allows more accurate assessment of the reproductive system [88]. In expert hands, US may depict the site of perforation, as an anterior hypo/anechogenic line, extending to the serosa; the uterus appears bulky and empty, while the placenta and fetal parts are located in the abdominal cavity. The ancillary non-specific findings of UR, such as concomitant hemoperitoneum and extrauterine hematoma, are the most commonly detected. On Color Doppler imaging, vascularity at the site of uterine breach may be decreased due to intra-myometrial hematoma [85,89,90]. When US in inconclusive, MRI represents a problem-solving tool with a larger FOV and improved visualization of the uterine wall, helping to diagnose antepartum UR (Figure 5).

On T2-weighted images, the placenta and amniotic sac are hyperintense, whereas the myometrium is isointense to muscle, allowing the detection of the tear itself. The high-contrast-tissue resolution of MRI provides for the depiction of fetal parts and extrauterine membranes. On T1-weighted images, hyperintensity of blood products identifies the hemoperitoneum [4,7,91].

Additionally, MRI is more accurate than US in differentiating UR from other uterine wall defects, such as UD, due to its ability to identify an intact overlying serosal layer. A correct differential diagnosis between UR and UD is crucial for setting up the most suitable treatment: UR requires an emergency caesarean section, whereas UD can be treated conservatively with antibiotics [7,81].

MRI allows for the detection and characterization of large bladder flap hematomas (>4 cm in size) associated with UR or more often with UD (Figure 6).

When they are abscessed, large hematomas may cause fever and abdominal pain: in these patients, cross-sectional imaging reveals a peripheral-enhancing purulent collection between the lower uterine segment and the bladder wall (Figure 7).

The use of MRI in emergencies is limited by availability in emergency departments and by the long scan times. According to the guidelines established by ACOG, the use of CT in pregnant patients should be considered when the benefit of the mother outweighs the potential harm from ionizing radiation [92]. CT is the modality of choice in postpartum patients and shows significantly higher sensitivity than other imaging modalities in the detection of pneumoperitoneum or abscesses associated with uterine perforation [8]. The tear is depicted as a hypodense myometrial defect in uterine layers, extending from the serosal surface to the myometrium. Multi-planar reconstructions may help in identifying the site and extent of the breach [93].

### 3.5. Postpartum Hemorrhage

Postpartum hemorrhage (PPH) is one of the leading cause of maternal morbidity and mortality, accounting for 8% of maternal deaths in developed countries and 20% of maternal deaths in developing regions [3,94]. Primary PPH occurs within the first 24 h after delivery and is the most common type of obstetric hemorrhage with an incidence of 4–6% of all deliveries, whereas secondary PPH develops between 24 h and 12 weeks after delivery [95]. Uterine atony, defined as the absence of uterine contraction after delivery, represents the main cause of primary PPH, accounting for 75–90% of cases [95,96]: the diagnosis is established during the physical exam immediately upon conclusion of an obstetric vaginal or caesarean delivery, with a poor role for imaging [97]. Other causes of primary PPH include delivery-related lacerations, coagulation diseases, PAS disorders, uterine rupture, and RPOC.

Secondary PPH complicates 0.2 to 1% of deliveries; the main cause is RPOC, accounting for 30% of cases [98]. Less-common causes of secondary PPH include uterine arteriovenous malformations (AVMs), endometritis, and subinvolution of the placental site.

Contrast-enhanced CT should be performed to detect and localize active bleeding that appears as extravasation of the contrast agent. Significant arterial bleeding can be identified on the arterial phase, while small arterial or venous oozing can be detected during the delayed phase (Figure 8) [97].

#### 3.5.1. Retained Products of Conception (RPOC)

RPOC is defined by intrauterine retention of residual placental or trophoblastic tissue, representing one of the most common causes of post-abortion or postpartum hemorrhage [99]. RPOC complicates 1% of all full-term pregnancies, occurring more frequently after second-trimester miscarriage or surgical pregnancy termination with a reported rate of 6%. In women who underwent medical abortions, a prevalence up to 15% is reported [100]. RPOC associated with PAS has increased, especially in patients who have achieved pregnancy through assistive reproductive technology [101]. The clinical symptoms of RPOC include vaginal bleeding, pelvic pain, and fever. Although an accurate and timely diagnosis of RPOC represents a challenge, it is crucial for guiding proper management. Transvaginal US is the first-line imaging modality and a useful diagnostic tool in differentiating RPOC from normal intrauterine lochia and clots [99]. The main US features of RPOC include a thickened endometrium (ranging from 8 to 13 mm) and a variable amount of heterogeneous and echogenic material within the endometrial cavity, sometimes presenting as a mass, with vascular flow at Color Doppler US [8]. Sellmeyer et al. observed that the detection of any vascularity in a thickened endometrial echo complex (EEC) or mass is likely to represent RPOC with a predictive positive value of 96%; however, the lack of vascularity at Color Doppler US does not exclude the diagnosis of RPOC [99]. Four vascularity patterns have been established, in which the degree of vascularity of the endometrial versus the myometrial component is compared in the same image section and classified as type 0, 1, 2, or 3. Type 0 vascularity, defined as no detectable vascularity in a thickened EEC or mass, may correspond to a blood clot or avascular RPOC. The vascular patterns 1–3 show an increasing flow of endometrium compared to myometrium, helping the clinical management of RPOC [8,99]. Type 3, defined as marked endometrial vascularity (higher than that of normal myometrium in the same US section), has a PPV of 100%; in these patients, flow can be so robust as to mimic a uterine arteriovenous malformation (AVM). However a mass in the puerperal uterus with vascularity on US is much more likely to represent RPOC than uterine AVM. AVMs can be considered when RPOC have been ruled out and Color Doppler US demonstrates serpiginous tubular vascular structures centered in the myometrium [102]. When US is inconclusive, CT or MRI should be performed. On MRI, RPOC appear as intracavitary uterine soft-tissue mass with variable T1 and T2 signal intensities, depending on the degree of bleeding and necrosis (Figure 9).

Post-contrast MRI is useful to assess the enhancement, which is also heterogeneous (and can be partial, complete, or delayed). Other features to take into account are a variable degree of associated myometrial thinning and obliteration of the junctional zone [8,103]. The aforementioned US and MRI findings can overlap with those of gestational trophoblastic disease (GTD), a broad spectrum of clinical and histopathological entities arising from uncontrolled growth of trophoblastic tissue and including hydatiform moles (complete and partial), invasive moles, choriocarcinoma, and placental site trophoblastic tumor [104]; therefore, clinical and lab context are crucial. Specifically; the serum β human chorionic gonadotropin (beta hCG) value is typically normal or low in women with RPOC and significantly increased in those with GTD [103]. GTD should be considered in the differential diagnosis of hemorrhagic conditions with a positive pregnancy test after a delivery, a miscarriage, or a query ectopic.

US plays a key role in patients with suspected GTD to exclude pregnancy as a cause of elevated hCG levels [105]. Once GTD is suspected, Doppler US is a valuable tool to confirm the diagnosis; however, MRI represents a problem-solving method in selected cases, especially when malignancies are suspected to evaluate the local extent of disease [106].

#### 3.5.2. Uterine Arteriovenous Malformation

Uterine AVMs are uncommon and underdiagnosed vascular disorders that may cause life-threatening postpartum vaginal bleeding. AVMs are broadly classified as congenital or acquired. Acquired uterine AVMs usually result from previous trauma, infection, gestational trophoblastic disease, endometriosis, or uterine surgery [8,107]. Clinically, puerperal patients present pelvic pain associated with vaginal bleeding, which can be heavy or irregular, requiring blood transfusion and/or emergency hysterectomy. Transvaginal US is commonly performed as the first-line imaging test representing a useful, non-invasive diagnostic modality. At US, uterine AVM in seen as an ill-defined mass that consists of multiple hypoechoic tubular or cystic areas within the myometrium with a normal adjacent endometrium [108]. Color Doppler US shows in those same areas low resistance, multidirectional, and high-velocity turbulent flow, confirming the vascular nature of these lesions. Doppler findings of uterine AVM may overlap with other conditions, including gestation trophoblastic disease (GTD); therefore, serum b-hCG testing is recommended to rule out GTD (negative with AVM, positive with GTD) [109,110].

When US is indeterminate, MRI or CT can reveal an ill-defined mass with multiple serpiginous and ectatic vessels in the myometrium and parametrium, connecting to the uterine arteries: this finding, clearly depicted on MR-angiography, corresponds to the hypervascular areas on Color Doppler US, representing a hallmark of uterine AVMs [8]. In congenital AVMs, several arterial feeders may be detected, with a more prominent nidus, whereas acquired types are typically supplied by a single intramural and hypertrophied feeding artery draining directly in a single parauterine vein [111]. Other MRI features of AVM include bulky uterus, focal disruption of the junctional zone, and flow-void artifacts (Figure 10).

Contrast-enhanced CT is the modality of choice for confirmation of the diagnosis in puerperal patients presenting to the emergency department with severe hemorrhage [8].

The definitive diagnosis is still made on the basis of conventional angiography, which remains the gold-standard imaging technique reserved for patients undergoing therapeutic embolization. Arteriography shows a nidus of vessels, supplied by dilated feeding uterine arteries, with high-flow vascular dynamics and early venous drainage [111].

Conservative treatment and observation have been suggested for stable patients with a clinical suspicion of AMV, whereas selective uterine arterial embolization (UAE) or hysterectomy is recommended only for unstable patients. Surgical treatment should be reserved for those women in which UAE is not feasible or contraindicated [112].

#### 3.5.3. Endometritis

Postpartum endometritis is defined as an infection of the decidua or uterine lining that can involve all the layers of the uterus. This complication is the most common cause of low-grade postpartum fever, occurring in 2–3% of vaginal births and in up to 28% of caesarean sections [113]. The likelihood of developing postpartum endometritis is doubled in patients who underwent caesarean delivery compared to those who deliver vaginally; the risk also increases without antibiotic prophylaxis and with prolonged labor or RPOC [114]. Signs other than fever include abnormal vaginal discharge, vaginal bleeding, uterine tenderness, persistent uterine enlargement, and leukocytosis [113]. Endometritis is diagnosed and managed clinically with little role for imaging, which is often used to exclude concurrent pathologies (RPOC, infected hematoma, abscess) or when conventional antibiotic therapy is unsuccessful. US findings are nonspecific, overlapping with expected postpartum changes: an enlarged uterus with thickened, heterogeneous endometrium containing intracavitary fluid or echogenic debris consistent with clot [115]. Echogenic foci with distal shadowing or ring-down artifacts may be observed in women with gas in the endometrial cavity, raising suspicion of infection. However, intracavitary gas may also be a normal finding in puerperal uterus [97]. Rule et al. (2018) showed that sonographic features such as a subserosal hypoechoic rim and endomyometrial junction indistinctness are helpful in distinguishing patients with postpartum endometritis from other complications [116]. The diagnosis cannot be made according to a single finding, but it is crucial to integrate imaging features into the clinical scenario [115].

CT and MRI can be employed to further rule out underlying entities such as an abscess or pelvic septic thrombophlebitis.

The CT findings of postpartum endometritis are also non-specific, including a thickened endometrium with diffuse wall enhancement and an enlarged endometrial cavity within fluid, air, or debris. MRI can also demonstrate similar findings to CT scans, with air depicted as signal void on T1- and T2-weighted images [117]. Contrast administration can be helpful in detecting parametrial inflammation and pelvic abscess. A pelvic abscess is shown as a thick-walled and well-defined fluid collection with rim enhancement. An air–fluid level within the abscess and/or fat stranding of the surrounding peritoneal fat, due to inflammatory changes, can also be observed [118].

The treatment of endometritis is broad-spectrum antibiotics administration with proper management for any associated RPOC, infected hematoma or pelvic abscess (Figure 11) [113].

## Figures and Tables

**Figure 1 diagnostics-13-02890-f001:**
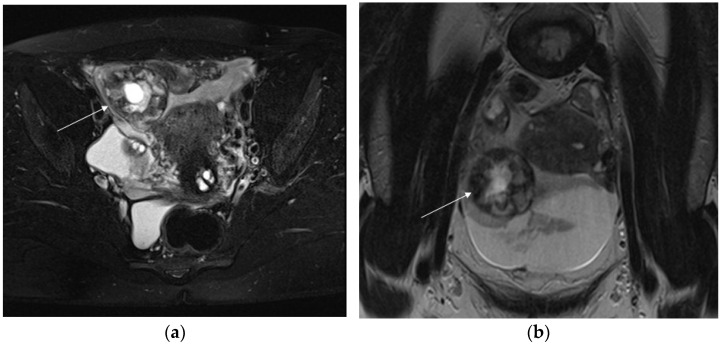
Axial T2-weighted fat-sat image (**a**) shows extrauterine gestational-sac-like thick-walled structure (arrow), exhibiting blood content and the typical “three ring” appearance. The inner and outer rings are thin and hypointense, whereas the middle ring is thick, displaying heterogeneous signal intensity due to small areas of hemorrhage. Coronal T2-weighted image (**b**) shows that gestational-sac-like structure (arrow) is separated from the right ovary. Pelvic effusion displays heterogeneous signal intensity with fluid–fluid levels for recurrent bleeding. Laparoscopic surgery was performed, confirming rupture of right ectopic tubal pregnancy.

**Figure 2 diagnostics-13-02890-f002:**
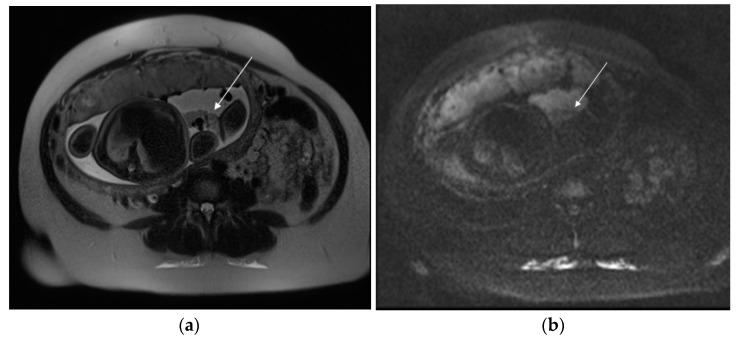
A 29-year-old patient at 35 weeks of gestation with acute pelvic pain and vaginal bleeding. Axial T2 (**a**) and DWI (**b**) sequences show a hematoma (arrow), contained within amnion and chorion and limited by reflection of amnion on placental insertion, suggestive of subamniotic placental abruption.

**Figure 3 diagnostics-13-02890-f003:**
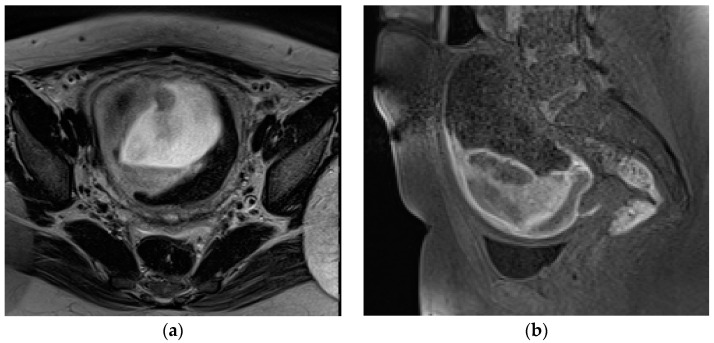
A 23-year-old patient at 12 weeks of gestation admitted to the emergency room for vaginal bleeding and acute pelvic pain. MRI detects large subchorionic placental abruption. Axial T2 Turbo-Spin-Echo (**a**), sagittal T1 fat-sat (**b**) sequences show heterogeneous signal of the hematoma, suggestive of the coexistence of acute and subacute hemorrhage.

**Figure 4 diagnostics-13-02890-f004:**
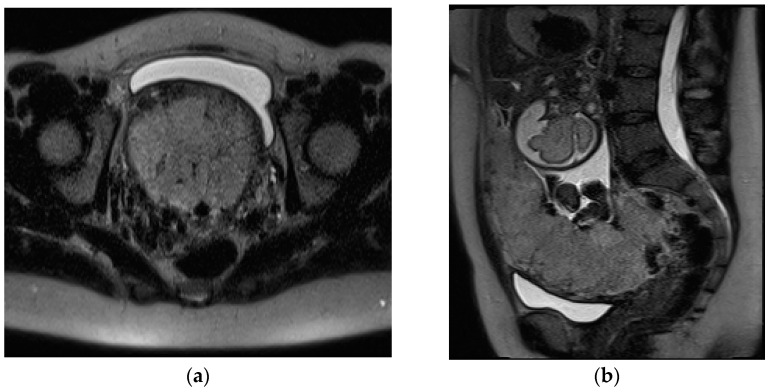
A 32-year-old pregnant patient at 28 weeks of gestation with acute pelvic pain. Axial (**a**) and sagittal T2-weighted images (**b**) demonstrate a heterogeneous placenta, with abnormal placental bulging, dark intraplacental bands, and interruption of the myometrium.

**Figure 5 diagnostics-13-02890-f005:**
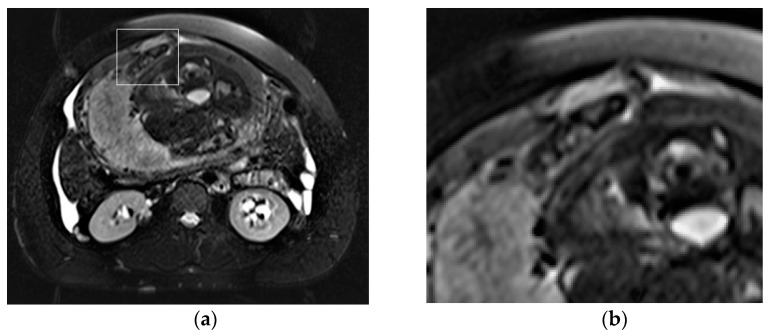
Acute abdominal pain in a 37-year-old patient at 34 weeks of gestation, with the suspicion of acute appendicitis. Axial T2-weighted fat-sat image (**a**) shows a full-thickness tear within the anterior uterine wall that also includes the uterine serosa, with herniation of the hand of the fetus through the defect, as highlighted by the square (**b**).

**Figure 6 diagnostics-13-02890-f006:**
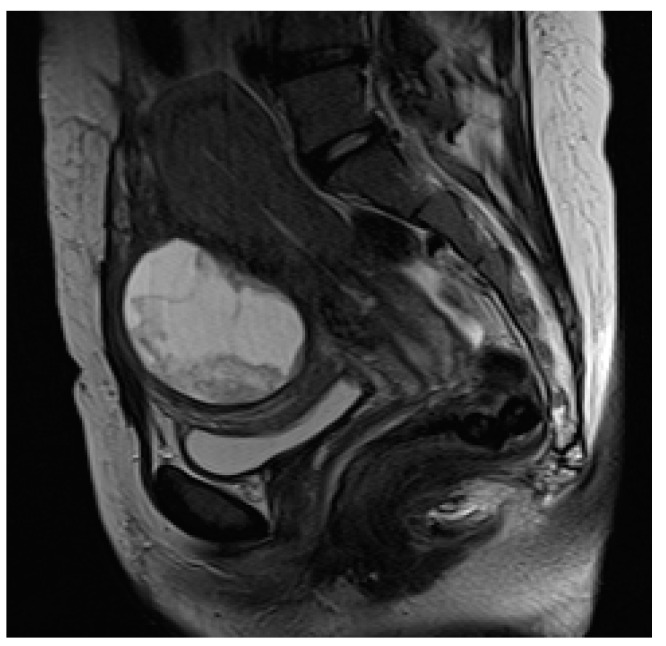
Puerperal patient, 34 years old, on the 14th day after caesarean section. Sagittal T2-weighted sequence shows large bladder flap hematoma (6.5 cm in maximum diameter), resulting from uterine dehiscence at the incision site and located between low uterine segment and bladder wall.

**Figure 7 diagnostics-13-02890-f007:**
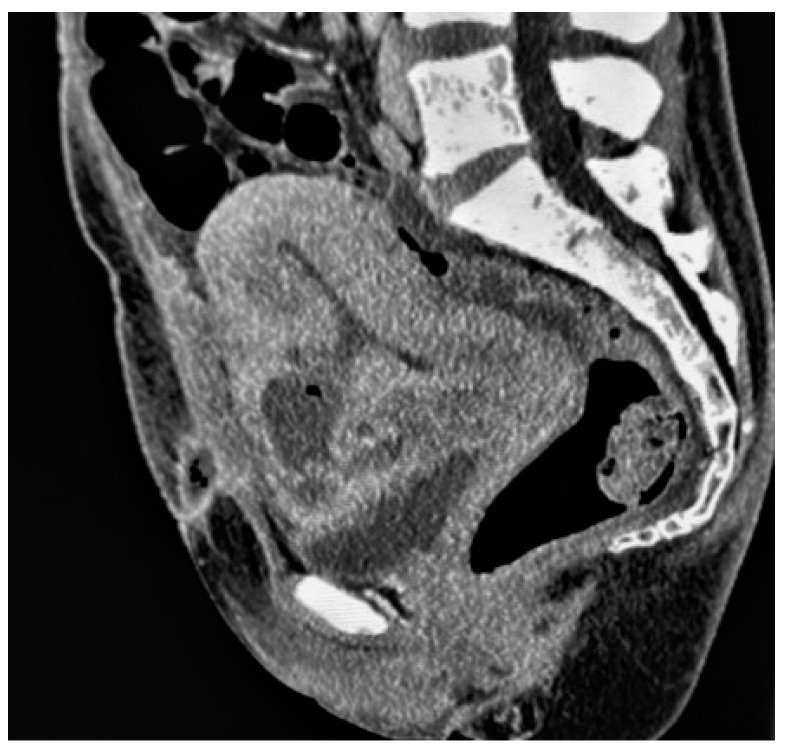
Bladder flap hematoma complicated by an abscess in a patient who had undergone caesarean delivery 8 days earlier, presenting with fever and hypogastric tenderness. Sagittal-reformatted contrast-enhanced CT image shows a rim-enhancing and gas-containing collection at the anterior lower incision site that communicates with the endometrial cavity.

**Figure 8 diagnostics-13-02890-f008:**
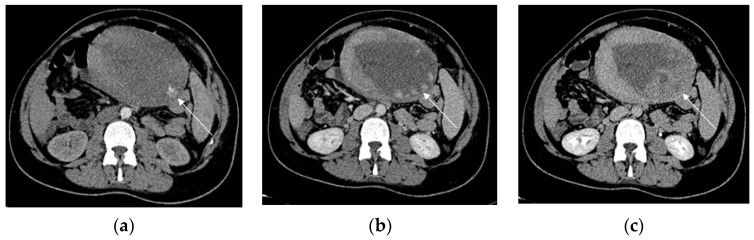
Primary postpartum hemorrhage occurring 6 h after delivery, in a 28-year-old patient with previously undiagnosed PAS disorder. Axial early arterial phase (**a**) contrast-enhanced CT scans demonstrate a subtle intrauterine contrast extravasation (arrow) in the left side of the endometrial cavity. The contrast extravasation is clearly depicted in the portal phase (**b**), whereas the delayed phase (**c**) demonstrates a more diffuse contrast extravasation in the same side.

**Figure 9 diagnostics-13-02890-f009:**
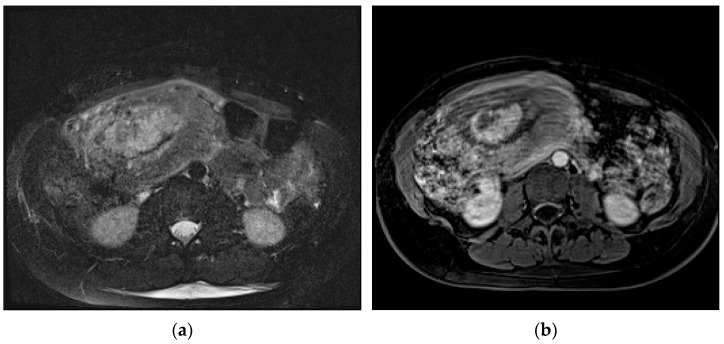
Postpartum hemorrhage occurring 11 days after delivery. Axial T2-weighted fat-sat sequence (**a**) shows hyperintense soft-tissue mass within the endometrial canal, briskly enhancing on T1-weighted post-contrast images (**b**). This findings were confirmed at surgery to represent components of retained placenta (RPOC).

**Figure 10 diagnostics-13-02890-f010:**
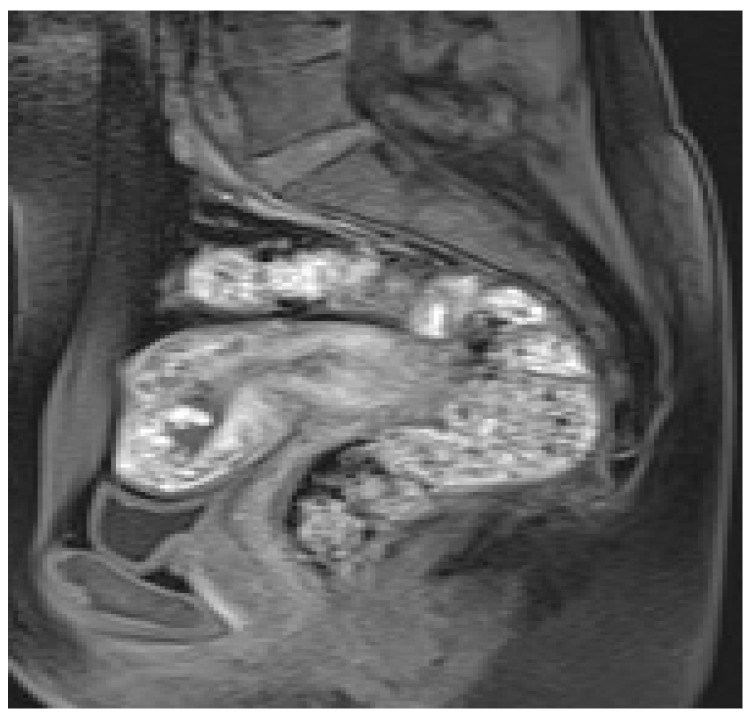
Uterine AVM in a 34-year-old patient who underwent uterine curettage after a miscarriage and presented to the emergency department with pelvic pain and vaginal bleeding. Sagittal T1-weighted contrast-enhanced image depicts in the fundus and anterior myometrium multiple serpiginous and ectatic vessels that enhance intensely. The myometrium has a bulky appearance.

**Figure 11 diagnostics-13-02890-f011:**
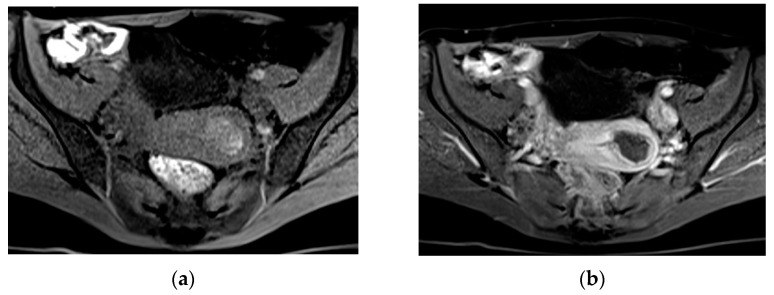
Endometritis in a 31-year-old patient admitted to the emergency department with fever and pelvic soreness 14 days after undergoing cesarean delivery. Axial T2-weighted fat-sat image (**a**) demonstrates dilated and fluid-filled endometrial cavity. After gadolinium administration (**b**), endometrium appears as thickened with heterogeneous enhancement.

## Data Availability

Data sharing is not applicable.

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
