# Peer review of "Imaging of Acute Abdominopelvic Pain in Pregnancy and Puerperium—Part I: Obstetric (Non-Fetal) Complications"

_diagnostics, 2023, doi:10.3390/diagnostics13182890_

Round 1

Reviewer 1 Report

Thanks to the author for summarizing the relevant literature on imaging examinations during pregnancy and puerperium for our clinicians. The article is fluent in writing and comprehensive in summarizing and evaluating the three kinds of imaging examinations during pregnancy and puerperium, which is helpful to guide clinical application. 

details

  • Incorrect spelling of Figure in line 288

  •  

Author Response

Response to Reviewer’s comments

Below our answers to Reviewers’ comments, we hope that our corrections will fulfill yours requests.

Thank you very much for the advices to improve our paper and for the critical analysis of our paper.

Best regards,

The authors.

Reviewer 1

Incorrect spelling of Figure in line 288. Thank you very much for pointing out this.

Corretion done ( highlighted in red).

Reviewer 2 Report

Dear Authors: The manuscript is complete and clear. Well done! Check if it is possible to include images as supplementary material and in a larger way.

Best regards

The reviewer

Author Response

Response to Reviewer’s comments

Below our answers to Reviewers’ comments, we hope that our corrections will fulfill yours requests.

Thank you very much for the advices to improve our paper and for the critical analysis of our paper.

Best regards,

The authors.

Reviewer 2

Check if it is possible to include images as supplementary material and in a larger way.

Thank you very much for this suggestion wich gave as the opportunity to improve our paper

We uploaded the images as pdf file as supplemental material.
